# The Predictive Value of Monocyte/High-Density Lipoprotein Ratio (MHR) and Positive Symptom Scores for Aggression in Patients with Schizophrenia

**DOI:** 10.3390/medicina59030503

**Published:** 2023-03-03

**Authors:** Ning Cheng, Huan Ma, Ke Zhang, Caiyi Zhang, Deqin Geng

**Affiliations:** 1Department of Psychiatry, First Clinical College, Xuzhou Medical University, Xuzhou 221000, China; 2Department of Psychiatry, The Affiliated Xuzhou Oriental Hospital of Xuzhou Medical University, Xuzhou 221000, China; 3Department of Medical Psychology, Second Clinical College, Xuzhou Medical University, Xuzhou 221000, China; 4The Key Lab of Psychiatry, Xuzhou Medical University, Xuzhou 221000, China; 5First Clinical Medicine College, Xuzhou Medical University, Xuzhou 221000, China; 6Department of Neurology, The Affiliated Hospital of Xuzhou Medical University, Xuzhou 221000, China

**Keywords:** schizophrenia, aggression, inflammation, high-density lipoprotein, monocyte, the positive symptom scores, MHR

## Abstract

*Background and Objectives*: Schizophrenia with aggression often has an inflammatory abnormality. The monocyte/high-density lipoprotein ratio (MHR), neutrophil/high-density lipoprotein ratio (NHR), platelet/high-density lipoprotein ratio (PHR) and lymphocyte/high-density lipoprotein ratio (LHR) have lately been examined as novel markers for the inflammatory response. The objective of this study was to assess the relationship between these new inflammatory biomarkers and aggression in schizophrenia patients. *Materials and Methods*: We enrolled 214 schizophrenia inpatients in our cross-sectional analysis. They were divided into the aggressive group (n = 94) and the non-aggressive group (n = 120) according to the Modified Overt Aggression Scale (MOAS). The severity of schizophrenia was assessed using the Positive and Negative Syndrome Scale (PANSS). The numbers of platelets (PLT), neutrophils (NEU), lymphocytes (LYM), monocytes (MON) and the high-density lipoprotein (HDL) content from subjects were recorded. The NHR, PHR, MHR and LHR were calculated. We analyzed the differences between those indexes in these two groups, and further searched for the correlation between inflammatory markers and aggression. *Results*: Patients with aggression had higher positive symptom scores (*p* = 0.002). The values of PLT, MON, MHR and PHR in the aggressive group were considerably higher (*p* < 0.05). The NHR (r = 0.289, *p* < 0.01), LHR (r = 0.213, *p* < 0.05) and MHR (r = 0.238, *p* < 0.05) values of aggressive schizophrenia patients were positively correlated with the total weighted scores of the MOAS. A higher MHR (β = 1.529, OR = 4.616, *p* = 0.026) and positive symptom scores (β = 0.071, OR = 1.047, *p* = 0.007) were significant predictors of aggression in schizophrenia patients. *Conclusions*: The MHR and the positive symptom scores may be predictors of aggressive behavior in schizophrenia patients. The MHR, a cheap and simple test, may be useful as a clinical tool for risk stratification, and it may direct doctors’ prevention and treatment plans in the course of ordinary clinical care.

## 1. Introduction

Schizophrenia, with a global prevalence of 1%, is a multi-system disease with no recognized cause, and is one of the fifteen leading causes of disability [1,2]. It accounts for about 12.3% of the global disease burden [3]. Schizophrenia is often demonstrated as a disorder of thought and behavior, including hallucinations, delusions, and disrupted speech and thought patterns, as well as cognitive symptoms such as deficits in working memory and cognitive flexibility [4]. It can lead to a deterioration of the patient’s social functioning, and places great stress on their families and on society as a whole [5]. In a nationwide study of violent behavior in schizophrenia patients conducted in the United States, 1410 patients underwent clinical and violent behavior evaluation. It was discovered that up to 19.1% of participants had engaged in violent behavior within the previous six months, with 3.6% of the participants exhibiting severe violent behavior [6]. The prevalence of aggressive behavior in hospitalized patients with schizophrenia in China ranged between 15.3% and 53.2% [7]. A study found that agitation, including aggression, is frequently the main or initial symptom of a patient receiving medical care or being admitted to the hospital [8]. According to research, about 1/5 patients hospitalized to acute psychiatric facilities may engage in violent behavior. Similar factors that contribute to violence in individual patients also contribute to levels of violence in psychiatric units (male gender, diagnosis of schizophrenia, substance use and lifetime history of violence) [9]. During acute episodes of schizophrenia, aggressive behavior is a common feature. As a result of their aggression, they may cause emergency events and tricky problems, including issues of legal liability and public security [10]. Hunter claimed that there were 973 beds in a forensic mental institution, and a total of 134 major injuries per year. The cautious estimate for the average cost per injury was $5719, for a total annual loss of $766,290 [11]. The health, safety and general wellbeing of patients, staff and others are endangered due to violent and aggressive behaviors, which cause substantial financial and medical costs. Therefore, more clinical indicators are needed to assess the risk of aggression in schizophrenia.

However, the etiology and mechanism of aggressive conduct are not completely understood at present. Studies from cat models found that inflammatory mediators are one of the reasons for aggressive behavior [12,13,14,15]. Furthermore, the application of proinflammatory proteins can boost defensive anger in cats [13]. In other animal models, aggressive behavior was induced by the injection of proinflammatory cytokines in the key brain areas [15]. Similarly, a growing body of studies have found that abnormal levels of inflammatory markers are associated with aggressive behavior, emotional management and cognitive difficulties in patients with mental disorders [16,17,18]. Moreover, Fanning et al. found that childhood trauma has a significant effect on violent and aggressive behavior in adult schizophrenia. There is long-lasting and low-grade inflammation following exposure to adversities in childhood. Immune system disruptions affect brain processes related to controlling aggressive behavior [19,20]. Immunologic dysfunction is also implicated as a primary factor in the schizophrenia pathomechanism, according to a growing body of solid evidence [21,22,23]. One of the hypotheses is that immune system abnormalities may be one of the etiologies for schizophrenia. Lymphocytes (LYM), neutrophils (NEU) and monocytes (MON) play significant roles in the inflammatory response [21]. Interestingly, there is growing evidence that suggests platelets (PLT) play a role in schizophrenia pathophysiology via the serotonin route and inflammation theory [24,25]. Increased levels of blood lymphocytes, monocytes, neutrophils and platelets, along with an increase in severity, were associated with the onset of aggressive behavior in patients with schizophrenia [23]. Aggressive conduct in schizophrenia has been shown to positively correlate with the neutrophil–lymphocyte ratio [26].

High-density lipoprotein (HDL) is one type of cholesterol. Due to HDL’s antioxidant properties, low-density lipoprotein (LDL), a risk factor for coronary heart disease, is protected from oxidation, decreasing or eliminating LDL’s ability to harm endothelial cells [27]. HDL has anti-inflammatory properties as well. Modified HDL inhibits the expression of adhesion molecules that are activated by cytokines, mediates cholesterol outflow from peripheral tissues, and eventually slows the progression of inflammation. The level of HDL fluctuates when inflammation occurs in the body, and it can prevent the activation of monocytes [28]. Thromboxane 2 secretion, fibrin linkage and platelet aggregation are all inhibited by HDL. HDL is a crucial part of lipid rafts. The lipid rafts are involved in cell signaling and cytoskeletal communication [29]. It has been shown that lipid rafts play an important role in neurodegenerative diseases [30]. The lipid rafts affected by cholesterol can change synaptic transmission and nerve plasticity, and can also affect the levels of serotonin and dopamine [29]. 5-hydroxyindoleacetic acid (5-HIAA), a central nervous system metabolite of serotonin, has been linked in studies to impulsive and violent behavior. 

Considering the roles that NEU, MON, LYM, PLT and HDL play in inflammation, PHR, MHR, LHR and NHR have recently been considered as newer inflammatory indicators. In various inflammatory illnesses, those indicators also were proposed as prospective markers of inflammatory responses and oxidative stress. Numerous studies found that the NHR and MHR were associated with the occurrence, development and prognosis of cardiovascular disease, ischemic stroke, cancer, erectile dysfunction, chronic kidney disease, Parkinson’s disease and so on [31,32,33,34]. The MHR, PHR, NHR and LHR are new types of inflammatory response markers that combine inflammation and anti-inflammation. Some scholars found the relevance of this class of inflammatory markers to psychiatric disorders. Such inflammatory markers have also been correlated with depressive disorders [35,36]. In 2021, Sahpolat et al. first found significant differences in the MHR between patients with schizophrenia and healthy populations in a small sample study [37]. In 2022, Yanyan Wei et al. recruited 13,329 patients with schizophrenia, 6005 patients with bipolar disorder and 5810 healthy people for a study, and subsequently found that the MHR could be used as a distinguishing factor between healthy people and SCZ patients [38]. Although it is true that there are relatively few inflammatory markers of this class in research on psychiatric disorders—which is in the exploratory phase—there are studies demonstrating the relationship of this class of indicators to psychiatric diseases.

Currently, numerous studies have evaluated the levels of the NHR, MHR, LHR and PHR in patients with mental disorders. However, the relationship between these ratios and aggression has been investigated less. In this study, we will investigate the relationship between the PHR, MHR, NHR and LHR and aggression in schizophrenia patients, similar to investigating the relationship between CRP, IL-6 and IL-10 and aggression. We predicted that these inflammatory indicators would be different in schizophrenia patients, with and without aggressive behavior. Therefore, the purpose of our study was to identify the indicators listed above that were significantly different between schizophrenia patients who displayed violence and those who did not. We also hoped to find potential biomarkers which could assess aggression in schizophrenia. Based on the results of previous studies, the following were our hypotheses: (1) Indicators of immunoinflammatory activity differ between the aggressive and non-aggressive groups; (2) the MHR, PHR, NHR or LHR may be predictive indicators of aggression in schizophrenia patients.

## 2. Materials and Methods

### 2.1. Participants

All of the schizophrenia inpatients were enrolled in The Affiliated Xuzhou Eastern Hospital of Xuzhou Medical University from January 2021 to June 2022 (n = 214). The following were the inclusion criteria: (1) inpatients between the ages of 18 to 65; (2) meeting the diagnostic criteria for schizophrenia based on the Diagnostic and Statistical Manual of Mental Disorders, Fifth Edition (DSM-5); (3) no psychiatric-related medications, immunosuppressive drugs or lipid-lowering drugs for at least three months. The exclusion criteria were as follows: (1) severe endocrine disorders, immunological disorders or physical illnesses (such as heart disease, diabetes, thyroid disease, inflammation and others); (2) pregnancy and the use of prescription weight-loss drugs or glucocorticoids, anti-arrhythmics or insulin; (3) people undergoing hypolipidemic treatment, or those who previously had hormonal problems (statins). Within 24 h of being admitted to the hospital, each patient consented to a mental evaluation using psychiatric scales, and an interview. For the patient group, the PANSS was employed to assess the severity of the schizophrenia [39]. The scale consists of the positive symptom, the negative symptom and the general psychopathology. The MOAS, including verbal aggression, aggression against property, auto-aggression and physical aggression, was used to assess the aggression [40]. The PANSS and MOAS were assessed in the time frame of all information, mainly about one week before admission. The aggressive behaviors were evaluated using the MOAS. According to previous studies [40], subjects with total weighted scores greater than five were assigned to the aggressive group. We also defined this as the occurrence of aggression. Subjects were split into two groups based on the MOAS’s evaluation: the aggressive group (n = 94) and the nonaggressive group (n = 120). Each participant’s medical history was examined to learn about their length of sickness, as well as prior and present attempts at suicide and aggressiveness. Additionally, the study controlled and monitored the impact of confounders such as gender, age, marriage and family history of psychosis. All participants gave informed consent after being told of the study’s aim and purpose. Demographic data, including gender, age, education, length of illness, family history of psychosis, marriage and body mass index (BMI) were collected. Blood samples from post-admission patients are taken the next morning. Blood sample processing was carried out with the Sysmex XN-1000 fully automated blood cell analyzer manufactured by Sysmex Japan. The test results included the number of platelets, neutrophils, lymphocytes and monocytes, and the level of HDL. A homogeneous enzymatic technique using polyethylene glycol and dextran sulfate was used to measure HDL cholesterol. The MHR was determined using the formula MHR = MON/HDL. The same method was used to calculate the LHR, NHR and PHR. 

### 2.2. Statistical Analysis

Statistical data processing was performed in the SPSS26.0 (SPSS Inc., Chicago, IL, USA) software package. For each of the investigated variables, the Shapiro–Wilk test was used to assess whether the distribution was normal. Continuous variables were provided as the median and interquartile range (IQR), and the Mann–Whitney U test was utilized for analysis because the majority of the data were non-normally distributed. The chi-square test was employed to examine classification comparisons in this study instead of Fisher’s exact test, because no cells had an anticipated number of values ≤ 5. The Spearman correlation in the aggressive group was used to analyze the blood indexes and aggression. A binary logistic regression model only contained variables with *p* < 0.05. The likelihood ratio was analyzed to choose the predictors of aggression. For all analyses, a two-sided *p* < 0.05 was regarded as statistically significant. 

### 2.3. Ethics Statement

Our investigation was carried out after the Helsinki Declaration. The Ethics Committee of the Affiliated Xuzhou Eastern Hospital of Xuzhou Medical University reviewed and approved this study (approval number: 2022011807). We appreciate everyone who committed to taking part in this study.

## 3. Results

### 3.1. Comparison of Demographical and Clinical Characteristics between Groups

This study included a total of 214 schizophrenia inpatients. Demographic data and clinical characteristics are shown in Table 1. The aggressive group consisted of 94 patients, including 55 males and 39 females. There was a statistically significant gender difference between the two groups (*p* = 0.028). The aggressive group of patients was disproportionately male. In comparison to the non-aggressive group, the aggressive group had a much higher percentage of males. The positive symptom scores (*p* = 0.002) significantly varied between the two groups about the clinical characteristics. Marriage (*p* = 0.433), age (*p* = 0.32), length of illness (*p* = 0.149), the total scores of the PANSS (*p* = 0.116), the negative symptom scores (*p* = 0.878), the general psychopathology scores (*p* = 0.350), the years of education (*p* = 0.593) and the family history of mental illness (*p* = 0.584) did not differ statistically significantly between the two groups. 

### 3.2. Comparison of Hematological Parameters in the Two Groups

The values of hematological parameters were from schizophrenia, with and without aggression, including HDL, NEU, PLT, LYM and MON, PHR, LHR, NHR as well as the MHR in Table 2. All of the aforementioned indices—aside from HDL, NEU, LYM, NHR, and LHR—revealed differences between the two groups.

### 3.3. Serum Concentration, (a) PLT, (b) MON, (c) MHR and (d) PHR in Schizophrenia Patients by Sex, with and without Aggression

We also compared the concentrations of PLT, MON, and the PHR and MHR in men and women between the two studied groups (Figure 1). In male patients, but not in females, the value of PLT was significant in the aggressive group and non-aggressive group (*p* = 0.015, Figure 1a). However, the significant difference in MON was only observed in female patients with schizophrenia. There was a higher level of MON in female schizophrenia patients with aggression, compared with the non-aggressive female group (*p* = 0.019, Figure 1b). Moreover, the mean value of the MHR was significantly higher in female patients with aggression in comparison to females with non-aggressive schizophrenia (*p* = 0.050, Figure 1c). In female patients with schizophrenia, it was found that aggressive patients had significantly higher values of the PHR (*p* = 0.041, Figure 1d).

### 3.4. Correlations among the NHR, LHR, MHR, PHR and the MOAS in the Aggressive Group

As shown in Table 3, in the aggressive group, we performed a Spearman correlation analysis of the values of the NHR, LHR, PHR and MHR and aggression. It showed the correlation between inflammation and aggression of different dimensions, including verbal aggression, aggression against property, auto-aggression and physical aggression. The value of the NHR was significantly positively correlated with total weighted scores (r = 0.289, *p* < 0.01) and auto-aggression (r = 0.319, *p* < 0.01). The LHR was positively correlated with total weighted scores (r = 0.213, *p* < 0.05) and physical aggression (r = 0.215, *p* < 0.05). The parameters of the PHR only positively correlated with auto-aggression (r = 0.227, *p* < 0.05). The MHR displayed a positive correlation with total weighted scores (r = 0.238, *p* < 0.01) and physical aggression (r = 0.230, *p* < 0.01).

### 3.5. The Related Factors for Aggression in Schizophrenia

In Table 4, we performed a logistic multivariate regression analysis. In this model, the MHR remained a significant predictor of aggression in patients with schizophrenia. The occurrence of aggressive behavior was used as the dependent variable. In binary logistic regression analysis, the covariates NHR, PHR, MHR, LHR, the positive symptom scores, and gender were all taken into consideration. The result which operated by likelihood ratio revealed that a higher MHR (β = 1.529, OR = 4.616, *p* < 0.05) and positive symptom scores (β = 0.071, OR = 1.047, *p* < 0.05) were significant predictors of aggression, whereas gender (*p* = 0.055), NHR (*p* = 0.692), LHR (*p* = 0.264) and PHR (*p* = 0.321) were not. 

## 4. Discussion

This study suggested that abnormal inflammation occurs in schizophrenia patients with aggression. All of the hypotheses were verified. Our analysis detected a significant increase in the values of MON, PLT, PHR and MHR in schizophrenia patients with aggression. This result is consistent with our hypothesis. The present study accesses the predictive values of the MHR for aggression in schizophrenia patients. This is consistent with our earlier theory. These results indicated that an elevated MHR and higher positive symptom scores were remarkably associated with aggression in schizophrenia patients. 

Neutrophils are the most abundant granulocyte type, accounting for 50% to 70% of all human leukocytes [41]. They serve as the body’s first line of defense against invasive diseases [42]. Additionally, neutrophils contribute to cell signaling and recruitment [43]. One study found that patients with higher neutrophil counts also had higher PLT and MON counts [44]. We found no distinctions between NEU, LYM, LHR and NHR in the two groups in the current investigation. We hypothesize that this is not only related to the traits associated with NEU and LYM, but also to the inflammatory hypothesis pathophysiology of schizophrenia. Although NEU are the body’s first line of defense, a number of cytokines control them. The majority of NEU are already produced, and have a short lifespan during the acute phase of inflammation [45]. Lymphocytes are the smallest type of white blood cells. They are mainly involved in the immune response process, including antibody production and cell-mediated immunity [46]. According to an earlier study, both psychiatric and neurodegenerative diseases have been linked to chronic inflammation of specific brain areas, which is defined by an infiltration of peripheral immune cells that may aggravate brain damage and lead to symptomatic clinical presentation [46]. Therefore, it seems reasonable that the results for NEU and LYM, which are often observed in the early phases of inflammation, are not significant in our study. As the largest white blood cells in the body, monocytes are crucial for the release of cytokines that are pro-inflammatory and pro-oxidant [47]. One of the key cells involved in chronic inflammatory conditions is the monocyte. A study found that anxiety, depression and hostile emotions promote monocyte accumulation [48]. In many physiological and pathological circumstances, activated platelets exhibit inflammatory properties, and can also control endothelial cell permeability [49]. Secondly, a major source of peripheral 5-hydroxytryptamine (5-HT) linked to aggressive behavior is platelets [50]. One study found that first-episode schizophrenia patients had considerably greater mid-term platelet levels than did healthy controls [51]. This is consistent with the results we observed. Our research adds support to the inflammatory theory of schizophrenia, and offers new perspectives on the relationship between inflammation and aggression, mood, and other behaviors.

A few studies revealed a link between aggression and inflammation, but no particular biological mechanism was shown [26,52,53,54]. A study reported that the attack score and concentration of 5-hydroxyindoleacetic acid in cerebrospinal fluid showed a strong negative connection, indicating that the severity of such violent behavior was inversely connected to 5-HT [55]. Inflammatory factors or inflammatory cells not only participate in the autoimmune reaction leading to brain dysfunction, but also decrease the activity of dopamine and 5-HT in the frontal lobe and hippocampus [56]. According to one study, oxidative stress may play a role in neuro-modulatory activity; it indirectly affects aggressiveness [57]. HDL-C has anti-inflammatory, anti-thrombotic, and antioxidative stress properties, in addition to improving lipid profiles [58]. Thus, the antioxidant function of HDL should not be ignored. Additionally, HDL can reduce inflammatory reactions and prevent native low-density lipoprotein (LDL) from oxidizing [59]. Inhibiting the cycle of inflammatory response processes and preventing monocytes from differentiating into macrophages are two effects of HDL-C. By limiting the growth of the progenitor cells that produce monocytes, HDL-C also works to counteract the pro-inflammatory and pro-oxidant actions of monocytes [58]. As a result, HDL has an indirect impact on many inflammatory cell types, in addition to its direct anti-inflammatory and antioxidant effects on the CNS. Low cholesterol levels have been linked to an increased risk of hostile, aggressive or suicidal conduct, according to numerous research studies [60,61]. Our findings contradict prior research, since we found no link between aggressiveness or impulsivity and HDL cholesterol levels. This could be as a result of the fact that we did not examine the connection between other cholesterol types, such as LDL and triglycerides, and aggression. Another reason for the above results may be that the influences of diet, habits and sports were not considered.

As for their complex relationship, composite indicators that include the NHR, LHR, MHR and PHR may be more trustworthy than a single measure in representing inflammatory levels. The MHR, PHR, NHR and LHR are a new group of inflammation markers that combine inflammation and anti-inflammation. They are regarded as basic, low-cost laboratory measures that can detect systemic inflammation in a variety of disorders. These indicators are inflammatory markers that are quick, simple, inexpensive and reproducible. They can highlight symptoms of systemic inflammation in a variety of illnesses, such as cardiovascular disease, ischemic stroke, cancer, erectile dysfunction, chronic kidney disease and Parkinson’s disease [31,32,33,34]. The NHR not only has strong predictive value for Parkinson’s disease, but is also closely related to disease duration [32]. The MHR, a novel inflammatory-oxidative stress biomarker, has been found by Ylmaz. et to be helpful in predicting the outcome of patients with primary nephrotic syndrome [62]. Significantly positive correlations between the MHR and LHR and cardiometabolic risk variables were observed [63]. Researchers also discovered that the PHR is a valid biomarker of nascent metabolic syndrome [64]. Abundant research indicates that those blood indexes can be regarded as independent predictors.

Similar to earlier research projects, our findings demonstrated that aggressive behavior in schizophrenia is associated with significantly higher PLT and MON counts [26,51]. Significant differences in the MHR and PHR were observed between the two groups. The NHR, MHR, PHR and LHR have been linked to schizophrenia severity in earlier research [37], but there was no evidence linking them to the likelihood of schizophrenic violence. According to the results of our study, patients who exhibited aggressive behavior had higher MHR and PHR values than the non-aggressive group. The MHR may be able to forecast when hostility will appear in schizophrenic individuals. This is the study’s most significant finding, since it suggests potential biomarkers for predicting aggression. It is important to note that the MHR correlation OR’s confidence interval in this study ranged from nearly 1.2 to 17.750, which suggests that the correlation may be anything from negligible to extremely significant. We speculate that the cause for this occurrence may be related to the modest variance in the MHR or the small sample size. To overcome this issue, we thought that future studies could increase the sample size, or that the MHR values could be examined after a logarithmic transformation, which would then enhance the model’s ability to forecast the future. In conclusion, the MHR may be a potential predictor of violence in schizophrenia patients, but many more research studies are needed to fully assess its predictive significance.

Prior studies reported a lot of different results about the association of inflammation and severity of psychiatric symptomatology in patients with mental illness. The study of injecting lipopolysaccharide into healthy volunteers showed that serum IL-6 increased in the peripheral immune system, and symptoms such as depression, anxiety and cognitive decline were caused [65]. Ranjit et al. found a positive association between hostility and the circulating levels of CRP and IL-6 [17]. IL-6 levels were found to be significantly higher in intermittent explosive disorder (IED) patients compared to controls, according to Emil et al. [54]. A different study on athletes found that when a game was coming up, football players’ rage triggered their immune system, leading to elevated IL-1 levels [66]. In 2017, Zhang et found that the hsCRP/IL-10 ratio was positively correlated with aggression [67]. There is a lack of consensus in public research on the molecular biomarkers of violent aggressiveness in schizophrenia patients. Even while some progress has been achieved, it is still only the beginning stages of exploration. At present, the mechanism of violent aggressive behavior in schizophrenia is still unclear. For the purposes of the research on inflammatory factors and their relationship, we presumptively believe that inflammatory factors not only have the ability to directly affect particular brain regions, but also to activate the distal neural circuit by triggering the primary triggers at a distance, which in turn trigger the chain reaction [68,69]. Since we cannot fully explain all violent behaviors in schizophrenic patients by high inflammatory activity in the body, we need to evaluate aggressive behavior comprehensively in the context of more situations.

Our findings demonstrated that gender disparities were significantly different, with more men among the aggressive subjects. According to many previous studies, men are more likely than women to engage in violent behavior [60,70,71]. It suggested that gender may play an important role in aggression. Aggression was observed more frequently in male patients with acute psychotic episodes, which is consistent with earlier findings [70,71]. First of all, we guess that those female patients tend to exhibit more frequent indirect aggression. They perceive the hostility and aggressiveness brought on by their insanity as a failure of self-control and self-preservation. Male patients, in contrast, may exhibit direct-physical aggression and verbal violence. Therefore, they are more likely to require physical restraints. Yet a study about borderline personality disorder showed that gender difference was not significant in aggression [72]. Some researchers found that girls engaged in more indirect aggression, such as cyber-aggression, when compared with boys [73]. This may imply that different genders have different forms of aggression. When preventing attacks, we should design unique strategies for each gender. Secondly, men and women have distinct hormone levels and thought processes, which may potentially have an effect on the occurrence of violent conduct. According to certain research, testosterone plays a complex role in social interaction in humans, and also promotes violence [74]. By changing the link between brain activity and the “threshold” for aggression, sex hormones promote persistent aggressive behavior [75]. However, in this study, the levels of hormones were not measured. Nevertheless, studies in Mongolian gerbils have shown that androgens can alter pro-social responses over both short- and long-term periods, promoting or inhibiting pro-social behavior depending on the social situation, but not affecting aggression [76]. 

Aggressive patients’ positive symptom scores were greater than those of non-aggressive patients in this study. This is consistent with the findings of previous studies [77,78,79]. More severe psychotic symptoms could be a mediator of greater violence. Based on these results, we can speculate that aggressive patients have more abundant and severe clinical symptoms. Aggression risk was higher in psychotic patients, particularly those with schizophrenia. The positive symptom scores were significant predictors of aggression, according to logistic regression. Therefore, we should be more cautious in evaluating the mood and aggressiveness of schizophrenia patients who present to the hospital with abundant clinical symptoms and more severe conditions. It can help us to better protect the personal safety of patients and healthcare workers. 

Medications such as clozapine and valproate have an impact on patients’ levels of aggression as well as on their blood lipids and blood cells [80,81,82]. Atypical antipsychotics, including clozapine, risperidone, quetiapine and ziprasidone, were widely believed to be the most effective medications for treating individuals with aggressive and violent behavior in the past [83,84]. At the moment, clozapine may be the only antipsychotic drug that effectively curbs violent behavior [80]. An increasing number of studies indicated that antipsychotic medicines’ multi-receptor mechanism of action frequently leads to metabolic problems in patients, including weight gain, hyperglycemia and hyperlipidemia [81]. In order to reduce the impact of confounding factors on this study, we made it a requirement for those enrolled to be off their medication for 3 months or more. The participants also had no previous history of dyslipidemia, and no history of statin use.

### Study Limitations and Future Prospects

This study has several limitations. Firstly, this was a cross-sectional design study. The causation could not be demonstrated, and a longitudinal study should be performed. A large number of studies are still needed to further validate our findings. Secondly, our subjects were not first-episode schizophrenia patients, and an analysis of the different immune inflammatory states still needs to be added. Additionally, we did not use a healthy control group of participants. Thirdly, other influencing factors were controlled, such as smoking, alcoholism, the levels of hormones, job, diet and exercise. Complex factors may contribute to aggressive behavior in schizophrenia, including substance abuse, gender, age, hallucination/delusional beliefs, poor impulse control, emotion, neurogenic trophic factor, hormone levels and so on [85]. In this study, we were not concerned with these affecting factors. Our study group was reduced because we excluded a subset of individuals with somatic disease, and concentrated on aggressiveness and immunological inflammation. One of the limitations of this study was the effects from prior medications (≥3 months) or other drugs. However, there are other forms of cholesterol which may also influence the inflammatory marker to HDL ratio, which were not included in our study. As a result, we view the findings of this study as a “potential prediction”, and numerous other future investigations are still required to assess their predictive validity.

We want to underline that neither a correlation nor a straightforward comparison indicated a causal connection between the values that were investigated. In conclusion, this study aimed to assess the relationship between inflammation and schizophrenia with aggression. Our research found that the level of inflammatory cells was different in schizophrenia with aggression or without. Further investigation revealed that the MHR and positive symptom scores were significant predictors of aggression. These results may help researchers identify variables that predict aggression in schizophrenia patients. The MHR can be quickly and affordably acquired. In the course of general clinical care, it can be utilized as a clinical tool for risk assessment, and can direct doctors in preventative and treatment planning. Our research adds support to the inflammatory theory of schizophrenia, and offers new perspectives on the relationship between inflammation and aggression, mood, and other behaviors.

## 5. Conclusions

In this study, the value of the MHR and the positive symptom scores may be predictors of aggressive behavior in schizophrenia patients. These preliminary results, however, require validation in sizable prospective investigations.

## Figures and Tables

**Figure 1 medicina-59-00503-f001:**
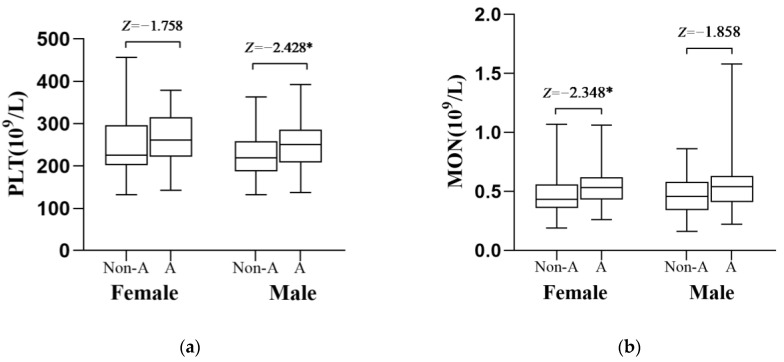
The values of MON, PLT, PHR and MHR in the aggressive group and non-aggressive group, separated by sex. The PLT was higher in the aggressive male group ((**a**), *Z*= −1.758, *p* = 0.015). Female schizophrenia patients with aggressive behavior had a higher level of MON ((**b**), *Z*=−2.348, *p* = 0.019). Higher values of the PHR ((**d**), *Z*=−2.045, *p* = 0.05) and MHR ((**c**), *Z*=−1.961, *p* = 0.041) were observed in the aggressive female patients. Non-A, non-aggressive group; A, aggressive group. PLT: platelets; MON: monocytes; PHR: platelet/high-density lipoprotein ratio; MHR: monocyte/high-density lipoprotein ratio. * *p* < 0.05.

**Table 1 medicina-59-00503-t001:** Demographical and clinical characteristics of schizophrenia inpatients with aggression and without.

Parameters	All Patients (n = 214)	Non-Aggressive (n = 120)	Aggressive (n = 94)	*p* Value
Gender, n (%)				0.028
Female	107 (50.00)	68 (56.70)	39 (41.20)	
Male	107 (50.00)	52 (43.30)	55 (51.40)	
Marriage, n (%)				0.433
Unmarried	96 (44.9)	51 (42.50)	45 (47.90)	
Married	118 (55.10)	69 (57.50)	49 (52.10)	
Age (years), median (IQR)	38.00 (18.00)	39.50 (18.00)	37 (17.00)	0.32
Family history of psychosis, n (%)				0.584
No	167 (78.00)	92 (76.70)	75 (79.80)	
Yes	47 (22.00)	28 (23.30)	19 (20.20)	
Education (years), median (IQR)	9.00 (3.00)	9.00 (3.00)	9.00 (6.00)	0.565
BMI (kg/m^2^), median (IQR)	23.88 (5.06)	24.01 (5.34)	23.62 (4.70)	0.664
Length of illness (years), median (IQR)	12.00 (11.00)	13.00 (10.00)	11.00 (9.00)	0.149
PANSS score, median (IQR)				
Total score	104.00 (29.25)	102.00 (27.00)	105.50 (32.00)	0.116
Positive symptom score	28.00 (8.00)	27.00 (7.00)	29.00 (7.00)	0.002
Negative symptom score	25.00 (14.00)	25.00 (14.00)	24.50 (14.00)	0.878
General psychopathology score	50.50 (14.00)	50.00 (12.00)	52.50 (17.00)	0.350

BMI, body mass index; PANSS, positive and negative syndrome scale; n, number of subjects in each group; IQR, interquartile range.

**Table 2 medicina-59-00503-t002:** Comparison of HDL, PLT, LYM, MON, LYM, NEU, PHR, NHR and MHR in the two groups.

	Non-Aggressive (n = 120)	Aggressive (n = 94)	Mann–Whitney U	
			Z	*p*
HDL (mmol/L)	1.180 (0.420)	1.145 (0.320)	−0.236	0.813
NEU (×10^9^/L)	3.795 (2.160)	4.165 (2.520)	−1.328	0.184
LYM (×10^9^/L)	1.815 (1.050)	1.990 (0.730)	−1.071	0.284
MON (×10^9^/L)	0.450 (0.220)	0.535 (0.200)	−3.088	0.002
PLT (×10^9^/L)	221.500 (80.000)	255.500 (82.000)	−2.731	0.006
NHR	3.336 (2.242)	3.458 (2.218)	−1.332	0.183
LHR	1.526 (1.255)	1.636 (0.890)	−0.888	0.375
MHR	0.394 (0.239)	0.419 (0.244)	−2.355	0.019
PHR	191.301 (118.245)	221.388 (110.130)	−2.234	0.025

PLT: platelets; NEU: neutrophils; LYM: lymphocyte; MON: monocytes; HDL: high-density lipoprotein; NHR: neutrophil/high-density lipoprotein ratio; PHR: platelet/high-density lipoprotein ratio; MHR: monocyte/high-density lipoprotein ratio; LHR: lymphocyte/high-density lipoprotein ratio; Values are represented as median (IQR). No significant differences in HDL, NEU, LYM, NHR or LHR were seen in our investigation. (*p >* 0.05). The monocyte counts in schizophrenia with aggression were significantly greater than in the non-aggressive group (*p* = 0.002). Furthermore, the levels of PLT showed the same result as MON (*p* = 0.006). Between these two groups, we compared four ratios, including the NHR, LHR, MHR and PHR. Both the MHR and PHR showed significant differences. Compared to the non-aggressive group, the aggressive group’s MHR value was significantly greater (*p* = 0.019). Furthermore, higher values of the PHR were observed in the aggressive group (*p* = 0.025).

**Table 3 medicina-59-00503-t003:** Correlations among the NHR, LHR, MHR, PHR and the MOAS in the aggressive group.

	Total Weighted Scores	Verbal Aggression	Aggression against Property	Auto-Aggression	Physical Aggression
NHR	0.289 **	−0.026	−0.001	0.319 **	0.199
LHR	0.213 *	−0.084	−0.036	0.134	0.215 *
PHR	0.115	−0.147	0.012	0.227 *	0.056
MHR	0.238 *	0.014	−0.055	0.163	0.230 *

NHR: neutrophil/high-density lipoprotein ratio; PHR: platelet/high-density lipoprotein ratio; MHR: monocyte/high-density lipoprotein ratio; LHR: lymphocyte/high-density lipoprotein ratio. Results given as Spearman correlation coefficient. * *p* < 0.05; ** *p* < 0.01.

**Table 4 medicina-59-00503-t004:** Multivariate analysis of aggressive behavior.

Aggression (Yes/No)					
Independent Variable	*β*	*SE*	Waldχ^2^	*p*	OR (95%CI)
MHR	1.529	0.687	4.953	0.026	4.616 (1.200, 17.750)
Positive symptom scores	0.071	0.026	7.236	0.007	1.074 (1.019, 1.131)

OR, odds ratio; SE, MHR: monocyte/high-density lipoprotein ratio.

## Data Availability

Data are available from the corresponding author upon request.

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
