# Peer review of "The Predictive Value of Monocyte/High-Density Lipoprotein Ratio (MHR) and Positive Symptom Scores for Aggression in Patients with Schizophrenia"

_medicina, 2023, doi:10.3390/medicina59030503_

Round 1
Reviewer 1 Report
The authors have appreciably tried to look into a biomarker for aggression in schizophrenia which is an important area of research however there are several issues which need clarification before any predictive value can be attributed to the given bio-marker.
Conceptual issues
Rationale: It seems that the rationale for this exploration of in-patient charts by the authors is based on a wide array of possible biomarkers being studied in Parkinson’s and in few studies on Schizophrenia. It should be stated clearly.
Hypothesis: Is the study exploratory? Even then it should have a hypothesis which forms the basis of doing these comparisons.
Inclusion and exclusion: The in-patient population reviewed consists of 214 markedly ill patients 9by high PANSS) who had a median 12 years of a psychiatric illness and with BMIs in the range of 23-24. Are we looking at a very stable well supported population as usually with such long term psychosis it’s hard to avoid metabolic syndrome with or without medications. There are no other parameters given such as HbA1c or other lipids to go by how healthy they were. In-fact diabetes is an exclusion which by itself would exclude a substantial portion of patients and is a limitation in assessing lipids as well as monocytes and thus the population is hardly representative of a general in-patient population and that should be stated unless authors have a better explanation.
One big confounder is that the authors did not account for any psychiatric/ or even medical comorbidity. Inflammatory markers have been linked to pretty much every psychiatric disorder, so how are these results only applicable to people with schizophrenia? If they are not, that should be stated.
Also any prior medications are also likely to have effects on lipids as well as inflammatory markers. If they were not accounted for it’s a limitation.
Measures:
It seems that the hospital was applying PANSS and MOAS at admission routinely (or were they applied by the study group retrospectively? Authors mentioned that the study was done retrospectively, hence my assumption). And thus, they capture 1-2 weeks prior to admission, or authors looked further in the charts for history of aggression? Did authors classify them as aggressive vs non-aggressive based on the reason for admission? Is not clear. Otherwise it would also be unclear if the lab changes are only likely associated with aggression, or there can be prior confounders such as long term effects of medications.
Authors have chosen a population off of any psychotropic for 3 months retrospectively. So the assumption is most likely this is a non-adherent and non-metabolic syndrome suffering population (with 12 years illness and now moderate to severe schizophrenia). So why was HDL checked in them? Or was it prospectively done as a part of the study? And if it was a routine in-patient admission procedure why only HDL and not other lipids and diabetes markers? Or the authors did not present them?
Otherwise as I mention further, the discussion section has assertions which are not fully supported by data unless clarified further.
Abstract:
Monocyte/HDL ratio appears twice in Abstract in the same sentence, kindly correct.
Introduction:
The authors have moved from asserting first that mental illness is associated with violence and then putting some references which are against this. There should be more sensitivity in putting such statements. Kindly see the specific comments below in this regard. The rationale for the study can still be developed even if authors do not go overemphasizing violence in patients with schizophrenia.
“Violence is one of the most common symptoms of mental illness” is not only not supported by evidence but is also a highly stigmatizing statement for sufferers of mental illness. Authors should avoid using such terms and give a reference to the actual prevalence if they feel a need to. It may be better to put statement like “A variable portion of in-patient subjects with schizophrenia exhibit violence ranging from X% to Y%.”. Some older European group reviews (Fazel et al, 2009) do mention this association which when controlled for substance use did not remain as high even for patients with Schizophrenia though remains significant.
No reference is given for “A study found…being admitted to the hospital”. The assertion further that Scz patients have 6 times higher rate…is not supported by the given review Hodgkins. In fact that review emphasizes to put more emphasis on prevention of aggression and recognizes that a major confounder is childhood conduct disorder which is indistinguishable before first psychotic break. Most the crimes had actually occurred before the psychotic break. Authors can keep statements to support ther rationale to do the study but must refrain from unnecessarily putting statements that paint a picture as if distressed patients with schizophrenia are just violent.
Avoid words such as “schizophrenics” and use a person centered statement than disease centered.
Studies from cat (or feline models)..reference not given. Even if I assume authors meant ref 14 as a reference for this statement that does not align with the plural “Studies”.
Sentence 101:seems incomplete “oxidative.” What?
Table 2, and Figure a, b which are based on Median (IQR) are confusing. Even if I just focus on the MHR in the table, the values 0.40 (0.24) and 0.42(0.24) do not appear to be significantly different. The authors may need to show them to more decimal points to show if on a larger scale these are different and may be clinically relevant. For eg. If a lab gets these two values like 0.40 and 0.42, should the lab raise suspicion in such a small range of 0.02, or is it that they had to look at more decimal places.
For the figure, I appreciate that authors at least gave a visual which many manuscripts do not do, but all the median (IQR) box-plots and stems seem to be overlapping, so I am puzzles how is the p value coming significant either the presentation of visual needs a change or some calculation. It may benefit from a statistical review.
Table 4:
The confidence interval of MHR related OR is interestingly quite wide ranging from being close to even 1.2 to 17.750. Such a wide confidence interval suggests that the correlation can as low as being non-significant to highly significant.
Discussion:
Line 266 “According to early studies, the subjects in our study…”, sentence is unclear, are they basing something on prior studies or contrasting something to.
The discussion further goes into studies like those on childhood trauma being pro-inflamatory, which do not seem relevant to the current study except for vaguely being related to “inflammation in psychiatric disorders.” They can form a rationale for the study but are redundant in discussion.
The authors can instead utilize the space more to discuss the limitations including why the results only “may predict” considering the numerical values they obtained as well as several limitations mentioned above related to the study design, inclusions and exclusions.
Reviewer 2 Report
1-It is a relatively new project to investigate the etiology of schizophrenia disorder and the role of inflammatory markers.
2-How comorbidity of substance (especially psychoactive drugs) use and schizophrenia is managed?
3-During the research period (from Jan.2021 to Jun.2022),How the fluctuations of the symptoms/Drug effect on aggression were managed?
4-During from Jan.2021 to Jun.2022,How the medication changes were managed?
5- In terms of ethics, please include acknowledgment .
